# What Lies behind Paraneoplastic Hypercalcemia Secondary to Well-Differentiated Neuroendocrine Neoplasms? A Systematic Review of the Literature

**DOI:** 10.3390/jpm12101553

**Published:** 2022-09-21

**Authors:** Elisa Giannetta, Franz Sesti, Roberta Modica, Erika Maria Grossrubatscher, Alberto Ragni, Isabella Zanata, Annamaria Colao, Antongiulio Faggiano

**Affiliations:** 1Department of Experimental Medicine, Sapienza University of Rome, 00161 Rome, Italy; 2Department of Clinical Medicine and Surgery, University “Federico II”, 80138 Naples, Italy; 3Endocrine Unit, ASST Grande Ospedale Metropolitano Niguarda, 20162 Milano, Italy; 4Endocrinology and Metabolic Diseases Unit, SS. Antonio e Biagio e Cesare Arrigo Hospital, 15121 Alessandria, Italy; 5Section of Endocrinology and Internal Medicine, Department of Medical Sciences, University of Ferrara, 44121 Ferrara, Italy; 6Endocrinology Unit, Department of Clinical and Molecular Medicine, Sant’Andrea Hospital, Sapienza University of Rome, 00189 Rome, Italy

**Keywords:** NET, PTHrP, PTH, NEN, 1,25-dihydroxyvitamin D, hyperparathyroidism, hypercalcemia

## Abstract

Background: Neuroendocrine neoplasms (NEN) originate from neuroendocrine cells ubiquitously spread throughout the body. Hypercalcemia associated with cancer is the most common life-threatening metabolic disorder in patients with advanced stage cancer. Paraneoplastic hypercalcemia is more commonly associated with hematological malignancies, renal and breast carcinomas, and squamous cell carcinomas, but it has also been described in patients with well-differentiated NEN, where it often remains undiagnosed. Among its causes, systemic secretion of parathyroid hormone-related protein (PTHrP) and ectopic production of 1,25-dihydroxyvitamin D and parathyroid hormone (PTH) may be considered paraneoplastic causes of hypercalcemia. In order to clarify the diagnostic work up of paraneoplastic hypercalcemia in patients with NEN, we perform a systematic review, which is lacking in the literature. Methods: We performed a data search using MEDLINE and SCOPUS including papers from 1961 to 2021. We selected articles on paraneoplastic hypercalcemia associated with well-differentiated NEN. Results: The search led to the selection of 78 publications for a total of 114 patients. Pooled data showed that the most frequent primary tumor site associated with paraneoplastic hypercalcemia was pancreatic NEN, followed by Pheochromocytoma. In most cases, paraneoplastic hypercalcemia was caused by PTHrP production and secretion. In more than two thirds of cases, paraneoplastic hypercalcemia was present at the time of NEN diagnosis and, in metachronous cases, was related to local recurrence, distant metastasis development, or tumor progression. In most patients, a combination of therapeutic approaches was employed, and reduction of the tumor burden was essential to control the paraneoplastic syndrome. Discussion: The onset of hypercalcemia associated with cancer in patients with well-differentiated NEN represents a major clinical challenge. The complex clinical and therapeutical management of paraneoplastic hypercalcemia implies the need for a multidisciplinary approach, aimed at controlling the clinical syndrome and tumor growth.

## 1. Introduction

Neuroendocrine neoplasms (NEN) originate from neuroendocrine cells which are distributed throughout the body. These tumors can synthesize and release biologically active substances such as hormones, peptides, or cytokines, causing distinct clinical syndromes [1] and differently impacting health-related quality of life [2,3,4]. Diagnostic and therapeutic management of functioning NEN is complex due to the high heterogeneity of these neoplasms in terms of clinical aggressiveness and the control of secretions. From the perspective of the precision medicine approach to NEN [5], the onset of paraneoplastic syndromes (PNS) should not be overlooked.

PNS are a heterogeneous group of clinical conditions, involving various systems, characterized by signs and symptoms occurring in association with malignancies. PNS are due to tumor-mediated production and release of different bioactive substances, or alternatively, by immune-mediated processes, and are not related to the specific organ or tissue from which they originate. PNS may occur before tumor diagnosis, concomitantly, or late in the course of clinical history and may influence therapeutic management. Consequently, PNS may impact prognosis and patients’ quality of life [6,7].

Hypercalcemia associated with cancer is the most common life-threatening metabolic disorder in patients with advanced-stage cancer. Hypercalcemia is associated with different neoplasms, may occur in up to 20–30% of all cancer patients, and is related to a poor prognosis. Thus, early diagnosis and intervention are of utmost importance in patients’ management. Hypercalcemia is more commonly associated with hematological malignancies, renal and breast carcinomas, and squamous cell carcinomas, but it has been also described in patients with well-differentiated NEN [8].

Hypercalcemia associated with cancer may be caused by: (i) systemic secretion of parathyroid hormone-related protein (PTHrP), a peptide produced by tumors with close homology in the N-terminal sequence to parathyroid hormone (PTH); (ii) osteolytic metastases, or, more rarely, by (iii) ectopic production of 1,25-dihydroxyvitamin D, which leads to intestinal hyperabsorption of calcium and increased osteoclastic bone reabsorption, and (iv) ectopic hyperparathyroidism [9].

Clinical presentation of hypercalcemia is influenced by its rapidity of onset and by its severity. Typical symptoms do not differ from benign hypercalcemia and may be nonspecific and develop gradually, leading to a delayed diagnosis. Signs and symptoms comprise gastrointestinal complaints such as nausea, vomiting, constipation, abdominal pain, and even anorexia, weight loss, bone pain, polyuria, weakness, and fatigue. Cardiovascular complications and arrythmias may also occur, as well as neurologic symptoms, especially in severe hypercalcemia (>14 mg/dl) [10].

Currently, data about NEN-related hypercalcemia come from case reports or case series. Most NEN-related hypercalcemia is secondary to the ectopic secretion of PTHrP, the so-called humoral hypercalcemia of malignancy, and it is more commonly described in association with pancreatic NEN (p-NEN). Different clinical presentations have been reported in association with different tumor stages, grades, and patient outcomes, as well as various therapeutic management strategies [11].

To provide a core of data about the epidemiology, clinical presentation, treatment, and impact on prognosis in patients with well-differentiated NEN-related hypercalcemia, we performed a systematic review [12].

## 2. Materials and Methods

We performed a systematic review of the literature according to the Cochrane Collaboration and PRISMA statement [13]. We searched for English-language articles in MEDLINE and SCOPUS, no timeframe restrictions were applied, including papers from 1961 to 2021. We searched for potentially relevant studies through these keywords: PTHrP AND NET/NEN; PTH AND NET/NEN; paraneoplastic hypercalcemia AND NET/NEN; and hypercalcemia AND NET/NEN. Eligibility criteria for study selection included studies on humans with any of the following designs: randomized clinical trials, prospective non-randomized trials, retrospective studies, case series, case reports, brief communications, and letters to the editor. We selected articles on paraneoplastic hypercalcemia associated with well-differentiated NEN, including paragangliomas, pheochromocytomas, medullary thyroid cancer, thymic and mediastinal, ovarian, uterine, cervical, gastroentheropancreatic, lung, and rectal NEN. For each paper, we analyzed patients’ age, sex, signs, symptoms, time presentation of hypercalcemia and hypercalcemia inducing molecules (PTH, PTHrP, 1,25(OH) vitamin D) or other peptide secretion. We furthermore evaluated the primary NEN’s site, grade, staging (with ENETS classification), type of metastasis at diagnosis of paraneoplastic hypercalcemia, NEN and hypercalcemia therapy, and patients’ survival from the onset of hypercalcemia. Each study was screened by abstract and title, and potentially eligible studies were further assessed in detail by retrieving full-length articles. Each full-length article was independently reviewed by three separate authors (AR, IZ, and FS) following the inclusion criteria. Three authors (AR, IZ, and FS) independently extracted data from the articles that met the inclusion criteria. A standardized form was used to extract relevant data.

Data are expressed as mean and standard deviation (SD) or median and 25–75% interquartile range (IQR), as appropriate. Normally distributed variables were assessed using the Shapiro–Wilk test. Homoscedasticity and homogeneity of variances were assessed by visual inspection and with Levene’s test. Differences between independent groups were evaluated using the t test for normally distributed variables and using the nonparametric Mann–Whitney test for non-normally distributed variables. Differences between the binomial proportions of independent groups of a dichotomous-dependent variable were assessed for homogeneity using the chi-square test or Fisher’s exact test, as appropriate. All statistical analyses were performed with SPSS Statistics version 27.0 (IBM SPSS Statistics Inc., Chicago, IL, USA).

## 3. Results

From the original number of 1281 studies, we excluded 1192 articles after title and abstract screening; reasons for exclusion included duplicates and studies in which hypercalcemia was due to primary hyperparathyroidism associated to genetic syndromes. We furthermore completed our research by analyzing the references of the selected papers (see Figure 1). We finally assessed 78 papers for a total of 114 patients for eligibility (see Table 1) [14,15,16,17,18,19,20,21,22,23,24,25,26,27,28,29,30,31,32,33,34,35,36,37,38,39,40,41,42,43,44,45,46,47,48,49,50,51,52,53,54,55,56,57,58,59,60,61,62,63,64,65,66,67,68,69,70,71,72,73,74,75,76,77,78,79,80,81,82,83,84,85,86,87,88,89,90,91]. The main clinical features of the gathered cases are summarized in Table 2.

The mean age of the patients was 46.3 ± 15.8 years and a slight majority of them was male (54.9%). The most frequent histological origin, with more than two thirds of the reported cases (72.8%), was p-NEN, followed by Pheochromocytoma (15.8%). All other NEN types were present only in a few patients. At the time of paraneoplastic hypercalcemia onset, most patients had a metastatic NEN disease (57.9%); in particular, the most common metastatic site by far was the liver, followed by the lymph nodes, bone, and lungs. Only 13.3% of patients with p-NEN (11/83) had a localized disease at paraneoplastic hypercalcemia onset, while the great majority of Pheochromocytoma cases (94.4%; 17/18) showed no sign of metastatic involvement at paraneoplastic hypercalcemia onset, although the adrenal tumors were, in the reports with available data, on average, quite large (mean size 5.5 × 6.6 cm).

### 3.1. Clinical Presentation

In 69.3% of cases, paraneoplastic hypercalcemia was present at the time of NEN diagnosis; this finding was especially true for those patients with Pheochromocytoma, among which 88.9% presented with paraneoplastic hypercalcemia at the time of tumor diagnosis. In the remaining cases, paraneoplastic hypercalcemia arose later in the course of the neoplastic disease, with a mean time from NEN diagnosis of 83.4 ± 56.3 months. The metachronous onset of paraneoplastic hypercalcemia was associated with the development of local recurrence or distant metastases and tumor progression in 43.8% and 50% of cases, respectively.

Among all cases, mean calcemic levels at paraneoplastic hypercalcemia onset were 14 ± 2.7 mg/dl. Calcemic levels in patients with p-NEN were higher than those in patients with a Pheochromocytoma, and this difference was found to be statistically significant (*p* < 0.001). No significant difference in the degree of hypercalcemia was found when comparing patients based on paraneoplastic hypercalcemia-producing molecules or paraneoplastic hypercalcemia onset (at NEN diagnosis vs. metachronous onset).

Data regarding the humoral factors responsible for paraneoplastic hypercalcemia were available only for 80 patients. In most cases (85%), PTHrP was considered the peptide implicated in paraneoplastic hypercalcemia onset and progression; PTH was elevated in 11.3% of patients with paraneoplastic hypercalcemia, while paraneoplastic hypercalcemia was driven by 1,25(OH) vitamin D in only three patients [14,17,26].

Besides hypercalcemia-producing molecules, 28.1% of the patients (mostly p-NEN) showed cosecretion of other peptides: the most frequent was calcitonin, followed by vasoactive intestinal peptide (VIP), pancreatic polypeptide, gastrin, somatostatin, and glucagon; there was a cosecretion of adrenocorticotropic hormone (ACTH) in only one case [88]. Interestingly, in seven cases, cosecretion of multiple peptides was reported [18,29,42,45,80].

### 3.2. Symptomatology

In 25.4% of cases, the clinical presentation of paraneoplastic hypercalcemia was not described; in the other 85 patients (74.6%), multiple signs and symptoms associated with paraneoplastic hypercalcemia were reported. Symptoms can develop gradually and become clinically evident only when blood calcium levels are very high. The severity of the onset depends not only on the age and the comorbidities of the patients, but also on the site of onset of the malignancy and on the grading of the primary NEN. It is very interesting to underline that the symptomatology of hypercalcemia could be synchronous with the diagnosis of the tumor or metachronous, and often correlated, with the progression of disease, even after many years.

The most recurring symptoms are anorexia and fatigue, which are described, respectively, in 37.6% and 31.8% of patients. For both symptoms, a progressive onset, often associated with other gastrointestinal symptoms, was reported. Anorexia is characterized by a typical gradual and involuntary weight loss, suggestive of neoplastic pathology. Among the typical symptoms of hypercalcemia, vomiting and nausea are described, respectively, in 24.7% and 21.2% of patients. Abdominal pain is another common clinical manifestation (21.2%) of paraneoplastic hypercalcemia, and it is depicted as an “indigestion pain” [20], an “abdominal discomfort” [34,55], and is associated with abdominal cramps without a well-defined localization. Constipation is the least frequent gastrointestinal symptom, complained about by only 9.4% of patients. This clinical picture is often evident at the diagnosis of NEN; however, it is difficult to discern with certainty whether it is caused by the neoplasm itself or by hypercalcemia. Regarding genitourinary manifestations of paraneoplastic hypercalcemia, synchronous polyuria and polydipsia are described, respectively, in 14.1% and 11.8% of patients and were linked to the onset of nephrogenic diabetes insipidus disease in one patient [74]. Paraneoplastic hypercalcemia, moreover, rarely causes dehydration up to the development of acute renal failure (2.4%); dehydration is often caused by diarrhea triggered by vasoactive hormones such as VIP [80,90]. The synchronous diagnosis of nephrolithiasis associated with NEN is mentioned in only four patients [32,45,63,81]. Neuropsychiatric symptoms are outlined in 7.1% of patients by the progressive development of cognitive dysfunction. Mental confusion at the diagnosis of NEN with an inability to maintain concentration and, in some cases, with an impaired short-term memory are described in 8.2% of patients. Hypercalcemia could, moreover, cause unexpected changes in patients’ behavior, anxiety, and depression up to the development of drowsiness, lethargy, and coma. Musculoskeletal symptoms are poorly described in the literature; however, muscle weakness is the most prevalent one (10.6%), followed by cramps, myopathy, and osteopenia and/or osteoporosis. With regard to bone pain (3.5%), Ataallah et al. and Rasbach er al. described two cases of arthralgia associated with hypercalcemia [16,78]. Cardiovascular manifestations are typically synchronous with the NEN diagnosis and include arrhythmias and hypertension, which are complained about by 14.1% of patients; however, in 10 of these patients, paraneoplastic hypercalcemia was caused by a Pheochromocytoma [35,61,73,76,77,81,89], thus is difficult to define if hypertension was caused by the effect of catecholamines or by paraneoplastic hypercalcemia. Lastly, Abraham et al. described a case report of paraneoplastic hypercalcemia correlated with diagnosis of p-NEN in a pregnant woman at 29 weeks’ gestation, which caused a symptomatology comparable with pre-eclampsia characterized by consciousness, headache, hypertension, and proteinuria [43].

### 3.3. Treatment Approach for Paraneoplastic Hypercalcemia

Only 85 cases had available data about the treatment used for the management of paraneoplastic hypercalcemia.

In most patients, a combination of therapeutic approaches was employed, mostly intravenous hydration, loop diuretics, and bisphosphonates (mainly pamidronate and zoledronate). In fewer cases, calcitonin and glucocorticoid were also employed, while the use of denosumab and cinacalcet was reported in only six and four cases, respectively. See Table 1.

Regarding antineoplastic therapy, data were available for 95 patients. In most cases, a combination of antineoplastic approaches was used. A total of 60 (63.2%) patients underwent surgery, both at the primary site and for metastatic or recurrent disease. Local techniques (embolization or radiofrequency ablation) were used in 17.9% patients for treating their liver metastases. Regarding medical therapy, somatostatin analogues (SSAs) were used in 37.9% of patients, chemotherapy was employed in 32.6% of patients, peptide receptor radionuclide therapy (PRRT) was administered in 10.5% patients, and target therapy with sunitinib and everolimus were both employed in five cases. See Table 1.

Data regarding paraneoplastic hypercalcemia response to therapy (both medical and antineoplastic) were available for 84 cases. Disease burden-reducing techniques (surgery, embolization) were able to control paraneoplastic hypercalcemia in 39.3% of cases, mainly in patients with localized, operable disease. Medical therapy alone could control paraneoplastic hypercalcemia in only 13.1% of patients, primarily through the utilization of bisphosphonates; in particular, intravenous hydration alone determined normalization of calcemic levels in only two patients and, when associated with other treatments, in three more cases. Medical antineoplastic treatments alone controlled paraneoplastic hypercalcemia in 20.2% of patients. In the remaining cases (27.4%), different combinations of therapies, including medical therapy for paraneoplastic hypercalcemia and antineoplastic (both surgical and medical), were used together to achieve paraneoplastic hypercalcemia control.

Survival data were available for 74 patients; median overall survival was 18 months (IQR range, 7–37). Among patients with p-NEN, median survival was 23.5 months (IQR range, 9.8–48).

## 4. Discussion

Nowadays, paraneoplastic hypercalcemia is a well-established paraneoplastic syndrome that is associated with many malignancies, even if the relationship between NEN and hypercalcemia is still little considered. Systematically reviewing the literature, we extracted that this rare condition was described in a total of 114 cases of patients with well-differentiated NEN.

The pancreas represents the most frequent localization of NEN associated with paraneoplastic hypercalcemia (72.8%), followed by Pheochromocytoma (15.8%). This observation is particularly interesting if we consider that, in an animal study, it has been demonstrated that PTHrP acts as a growth factor for pancreatic beta-cells [92] and that, in chronic pancreatitis, PTHrP functions as a mediator of proinflammatory and profibrotic cytokines, which in turn regulate PTHrP expression [93]. As in other malignancies, paraneoplastic hypercalcemia may also occur in NEN through several different mechanisms, including PTHrP secretion, PTH secretion, and calcitriol overproduction. In NEN, as in all other types of solid cancers, the most common cause of paraneoplastic hypercalcemia is the tumor production and release of PTHrP (85% of cases). PTHrP carries out a physiologic role in embryologic development and in mammary gland function, but it has no other known functional role in the adult metabolism [94]. PTHrP shares its amino acid sequence homology with PTH at its N-terminus and activates the type 1 PTH receptor, but it is encoded by a different gene [95]. Like PTH, PTHrP also increases calcium reabsorption in the kidney and stimulates osteoblasts to secrete receptor activators of nuclear factor-B ligands (RANKL), which bind to the RANK receptor on osteoclasts [96,97]. This interaction mediates the differentiation of osteoclast precursors into mature osteoclasts and increases bone resorption by osteoclasts. Since the most frequent cause of paraneoplastic hypercalcemia in NEN is PTHrP secretion, PTHrP levels should be checked in all patients with this clinical and biochemical suspect. The accuracy and reliability of laboratory assays for PTHrP have improved because of newer double-antibody techniques. Furthermore, when elevated at tumor diagnosis, PTHrP can be used as a biomarker to assess treatment response to therapy.

A less common cause of paraneoplastic hypercalcemia is the paraneoplastic ectopic secretion of PTH by tumors, which has been described in association with several malignancies, most of which are of the lung [98,99,100]. Only three cases of well-differentiated NEN associated with paraneoplastic hypercalcemia secondary to tumor-mediated overproduction of calcitriol are described in the literature [14,17,26]. Over production of calcitriol is a typical cause of paraneoplastic hypercalcemia in lymphomas [101,102], in which tumor cells or surrounding lymphocytes overexpress 1α-hydroxylase, which causes ectopic conversion of 25 hydroxyvitamin D to 1,25-dihydroxyvitamin D [103,104]. Calcitriol-related hypercalcemia derives from both increased intestinal and bone reabsorption of calcium.

This review showed the timing of paraneoplastic hypercalcemia occurrence during the “natural history” of well-differentiated NEN disease. In most cases, hypercalcemia is already present at diagnosis (69.3% of cases); in others, it develops during the disease (mean time from NEN diagnosis of 83.4 ± 56.3 months). In most cases, the metachronous occurrence of hypercalcemia is associated with disease progression/relapse. In the case of Pheochromocytoma, hypercalcemia was already present at diagnosis in 88.9% of cases, while in two cases, hypercalcemia was observed at the recurrence of the disease 94 and 204 months after the first surgical treatment, respectively. Hypercalcemia is almost always associated with the presence of distant metastases, except in 10.5% of cases, in which there were no metastases or there were only metastases to the local regional lymph nodes. Therefore, it seems that the tumor burden at diagnosis or during disease progression determines the capacity of hormone secretion by the neoplasm, which is different to what is described in the literature for Pheochromocytoma.

Paraneoplastic hypercalcemia is typically associated with severe clinical signs and symptoms and is often an oncologic emergency [9], while the paraneoplastic hypercalcemia of the NEN as a whole seems to give more moderate symptoms, similar to those of primary hyperparathyroidism. The most frequent symptoms are asthenia, gastrointestinal, and genitourinary disturbances. Severe symptoms such as pre-eclampsia, coma, lethargy, and arrhythmia have been described in extremely rare cases.

The management of paraneoplastic hypercalcemia in well-differentiated NEN is challenging. In fact, in our review, we observed that, in 27.4% of cases, the combination of multiple treatments (medical therapy for paraneoplastic hypercalcemia and different antineoplastic (both surgical and medical) treatments, variously combined) was required to obtain control of hypercalcemia. Disease burden-reducing techniques (surgery, embolization) were able to control paraneoplastic hypercalcemia, mainly in patients with localized, operable disease. Medical therapy alone could control paraneoplastic hypercalcemia in selected patients, primarily through the employment of bisphosphonates; in particular, intravenous hydration alone determined calcemic normalization in only two patients and, when associated with other treatments, the improvement is measured in very few cases. Medical antineoplastic treatments alone controlled paraneoplastic hypercalcemia in less than a quarter of reported patients. Therefore, the pooled data from our systematic review show that tumor debulking plays a key role in controlling paraneoplastic hypercalcemia in patients with well-differentiated NEN, so surgical treatment should be indicated whenever feasible. PRRT was administered in a limited number of cases; however, given its capability to control functioning tumors [105] and its potential role as a neoadjuvant therapy [106,107,108], PRRT could be prescribed either before surgery or in patients with progressive metastatic inoperable disease to reduce tumor secretion and tumor burden.

The onset of hypercalcemia associated with cancer in patients with well-differentiated NEN represents a major clinical challenge. Prior to the diagnosis of paraneoplastic hypercalcemia, physicians should rule out multiple endocrine neoplasia (MEN) 1 and 2 [109,110], in which the hypercalcemia could be due to primary hyperparathyroidism. Paraneoplastic hypercalcemia caused by ectopic production of PTH, although uncommon, should be considered in patients with p-NET when PTH levels are significantly elevated and there is no evidence of a parathyroid-related cause. Recognizing the association between elevated PTH levels and paraneoplastic hypercalcemia can prevent unnecessary parathyroid or exploratory neck surgery. Since paraneoplastic hypercalcemia must be recognized and framed promptly, and as it often remains undiagnosed, the complex clinical and therapeutical management of paraneoplastic hypercalcemia implies the need for a multidisciplinary approach, aimed at controlling the clinical syndrome and tumor growth. With the present review we have shown how paraneoplastic hypercalcemia in well-differentiated NEN was diagnosed and managed over the years and how important it is to conduct a personalized diagnostic and therapeutic process that provides an overview of the patient and his status.

In summary, compared with paraneoplastic hypercalcemia of solid and hematological tumors, paraneoplastic hypercalcemia in NEN shares PTHrP as the most common causal agent with solid tumors, while paraneoplastic hypercalcemia in lymphomas is more frequently caused by 1,25(OH) vitamin D. Furthermore, paraneoplastic hypercalcemia in NEN seems to be less severe than in solid and hematological tumors. Finally, the prognosis of paraneoplastic hypercalcemia of solid and hematological tumors seems to be worse than in NEN; this could be related to the milder symptomatology of NEN patients and to their better oncological prognosis.

A limitation of this review is represented by the difficulty of bibliographic research and data extraction. Since paraneoplastic hypercalcemia in NEN is a rare condition, we decided not to place timeframe restrictions in the selection of the articles; this allowed us to include a considerable number of cases in the review. However, it led to a lot of missing data, especially from the oldest articles. Indeed, relevant information such as grading, staging, and the paraneoplastic hypercalcemia-inducing molecule were not reported in some older case reports; interestingly, in cases from 1961 to 1991, an unspecified PTH-like substance was considered responsible for paraneoplastic hypercalcemia. Given that PTHrP was first isolated in 1987, we could speculate that, in those patients, PTHrP was the paraneoplastic hypercalcemia-driving molecule. See Table 1.

## Figures and Tables

**Figure 1 jpm-12-01553-f001:**
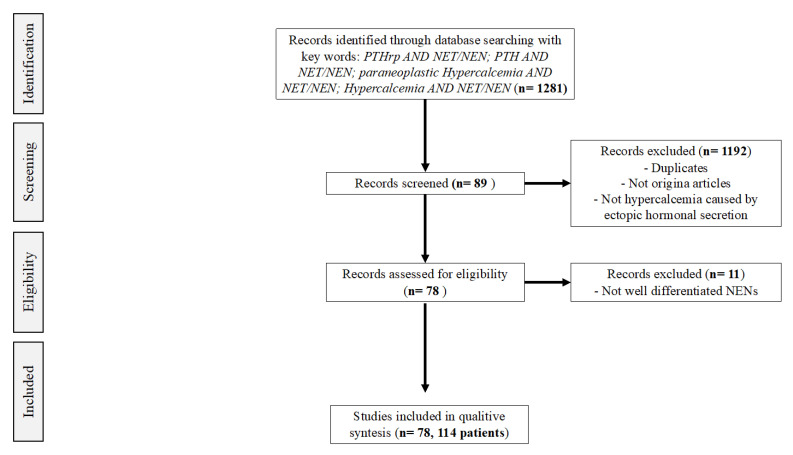
Flowchart of the literature search for the systematic review study. From: Liberati A, Altman DG, Tetzlaff J, Mulrow C, et al. (2009) The PRISMA Statement for Reporting Systematic Reviews and Meta-Analyses of Studies That Evaluate Health Care Interventions: Explanation and Elaboration. PLoS Med 6(7): e1000100. doi:10.1371/journal.pmed.1000100.

**Table 1 jpm-12-01553-t001:** Summary of cases of paraneoplastic hypercalcemia in well-differentiated NEN reported in the literature.

Author, Year	N° of Cases	Sex, Age (y)	Primary Site of NEN	Grade, Ki67	Initial Staging (ENETS)	Metastasis at Diagnosis of Paraneoplastic Hypercalcemia	Time to Onset of Paraneoplastic Hypercalcemia (Months)	Paraneoplastic Hypercalcemia-Inducing Molecules	Other Peptide Secretion	Paraneoplastic Hypercalcemia Therapy	NEN Therapy	Paraneoplastic Hypercalcemia Responsive to	Survival from Onset of Paraneoplastic Hypercalcemia (Months)
Giannetta E et al., 2021 [14]	4	M, 40	Pancreatic	G2, Ki67 = 5%	IV	Liver	At diagnosis	PTHrP	Calcitonin	Denosumab	SSA PRRT	Denosumab	36
		M, 45	Pancreatic	G2, Ki67 = 5%	II	-	96	NA	-	IV hydration Loop diuretics Cinacalcet Biphosphonate Denosumab Corticosteroids	Surgery SSA Everolimus PRRT	Corticosteroids	60
		F, 49	Pancreatic	G1, Ki67 < 1%	III	-	At diagnosis	1,25(OH) vitamin D	-	Hemodyalisis Biphosphonate Calcitonin	TAE Surgery	Surgery	156
		M, 69	Pulmonary	Atypical carcinoid, Ki67 = 9%	IV	Liver	At diagnosis	PTHrP	Calcitonin	Biphosphonate	Surgery	Surgery	NA
Copur MS et al., 2020 [15]	1	F, 62	Pancreatic	G3, Ki67 = 30%	IV	Liver	At diagnosis	PTHrP	-	Biphosphonate IV hydration	SSA 5-FU Oxaliplatin Surgery Pembrolizumab	Biphosphonate IV hydration	6
Ataallah B et al., 2020 [16]	1	F, 22	Pancreatic	NA	IV	Liver	At diagnosis	NA	VIP	Biphosphonate IV hydration	-	Biphosphonate	NA
Van Lierop AH et al., 2019 [17]	1	M, 50	Pancreatic	G2, Ki67 = 10%	IV	Spleen Liver	94	1,25(OH) vitamin D	-	Biphosphonate IV hydration	Surgery CapTem Nivolumab	Surgery	24
Gild ML et al., 2018 [18]	1	M, 47	Pancreatic	G1, Ki67 < 1%	NA	Lymphonodes	At diagnosis	NA	Glucagon PP	Biphosphonate Denosumab	SSA Surgery	Surgery	27
Daskalakis K et al., 2018 [19]	1	NA	Pancreatic	NA	IV	Liver	NA	PTHrP	-	Bisphosphonate IV hydration Cinacalcet	Surgery SSA Streptzotocine 5-FU IFNα PRRT Bevacizumab CapTem TAE Everolimus Sunitinib	TAE CapTem	NA
Symington M et al., 2017 [20]	1	F, 54	Pancreatic	G1, Ki67 < 1%	IV	Liver	At diagnosis	PTHrP	Gastrin	Bisphosphonate IV hydration	SSA	SSA	3
Lu C et al., 2017 [21]	1	M, 65	Mediastinal	Typical carcinoid	NA	-	At diagnosis	PTH	-	-	Surgery	Surgery	NA
Ranade R et al., 2017 [22]	1	M, 49	Pancreatic	G2, Ki67 = 12%	IV	Liver Bone	At diagnosis	PTHrP	-	NA	SSA PRRT CapTem	PRRT CapTem	21
Valdes-Socin H et al., 2016 [23]	1	M, 52	Pancreatic	G1, Ki67 = 2%	IV	Liver Spleen	At diagnosis	-	Calcitonin	Bisphosphonate Calcitonin IV hydration Cinacalcet	Streptozotocine Adriamycin FOLFOX SSA Sunitinib	Cinacalcet	48
Iliuta IA et al., 2015 [24]	1	M, 48	Pancreatic	NA	IV	Liver	At diagnosis	PTHrP	-	Bisphosphonate IV hydration Calcitonin Corticosteroids	SSA Everolimus PRRT	PRRT	NA
Teng J et al., 2014 [25]	1	M, 38	Pancreatic	G1, Ki67 < 2%	IV	Liver	At diagnosis	PTHrP	-	Hemodialysis Bisphosphonate Calcitonin IV hydration Denosumab Corticosteroids	Carboplatin Etoposide PRRT	Bisphosphonate Denosumab PRRT	22
Zhu V et al., 2014 [26]	1	F, 43	Pancreatic	G1-2, Ki67 = 2–5%	IV	Liver	96	1,25(OH) vitamin D	-	Bisphosphonate IV hydration Denosumab Calcitonin	SSA HACE Sunitinib CapTem	CapTem	24
Kamp K et al., 2014 [27]	9	M, 41	Pancreatic	NA	IV	Liver	At diagnosis	PTHrP	-	NA	NA	NA	NA
		M, 58	Pancreatic	G1	IV	Liver Lymphnodes	NA	PTHrP	-	NA	NA	NA	NA
		F, 40	Pancreatic	NA	IV	Liver Lymphnodes	At diagnosis	PTHrP	VIP	NA	NA	NA	NA
		M, 61	Pancreatic	G1	IV	Liver	At diagnosis	PTHrP	-	Biphosphonates IV hydration Denosumab	SSA PRRT Sunitinib	SSA PRRT	NA
		M, 60	Unknown	G2	IV	Liver	At diagnosis	PTHrP	-	NA	NA	NA	NA
		M, 38	Pancreatic	G1	IV	Liver	At diagnosis	PTHrP	-	NA	NA	NA	NA
		F, 42	Pancreatic	NA	IV	Liver Lymphnodes	At diagnosis	PTHrP	-	NA	NA	NA	NA
		F, 51	Pancreatic	NA	IV	Liver	At diagnosis	PTHrP	-	NA	NA	NA	NA
		F, 49	Pancreatic	G1	IV	Liver Lymphnodes	NA	PTHrP	-	Biphosphonates Corticosteroids IV hydration Loop diuretics	Surgery SSA PRRT TAE	-	18
Rossi RE et al., 2014 [28]	1	F, 25	Pancreatic	G2, Ki67 = 5%	IIA	-	At diagnosis	PTHrP	-	Biphosphonates IV hydration	Surgery TAE OLT Streptozotocin 5-FU Cisplatin	Surgery	192
Milanesi A et al., 2013 [29]	5	F, 49	Pancreatic	G1, Ki67 < 2%	IV	Liver	18	PTHrP	Somatostatin PP	Biphosphonates	SSA PRRT	Biphosphonates	36
		M, 53	Pancreatic	G2, Ki67 = 5–10%	IV	Liver	108	PTHrP	-	Biphosphonates IV hydration	Surgery SSA HACE CapTem	HACE Biphosphonates CapTem	96
		M, 52	Pancreatic	G2, Ki67 < 5%	IIB	Liver	48	PTHrP	Glucagon	Biphosphonates	Surgery HACE	Biphosphonates HACE	60
		F, 40	Unknown	G2, Ki67 = 10%	IV	Liver Lung	2	PTHrP	Gastrin	Biphosphonates IV hydration	Sunitinib Everolimus Carboplatin Etoposide	-	17, DOD
		M, 54	Pancreatic	G1, Ki67 = 2%	IV	Liver Bone	At diagnosis	PTHrP	-	Biphosphonates Calcitonin	PRRT SSA CapTem	CapTem	15
Shah RH et al., 2013 [30]	1	F, 53	Pancreatic	G1, Ki67 < 1%	IV	Liver Adrenal gland	At diagnosis	PTHrP	PP	Biphosphonates	SSA	Biphosphonates SSA	NA
Kanakis G et al., [31]	1	M, 58	Pancreatic	G2, Ki67 = 4%	IV	Liver	48	PTHrP	PP	Biphosphonates IV hydration Cinacalcet Glucocorticoids	Streptozotocine 5-FU IFNα SSA PRRT CapTem HACE Bevacizumab Everolimus	HACE Combined chemotherapy	72
Kandil E et al., 2011 [32]	1	F, 73	Neck	NA	NA	-	At diagnosis	PTH	-	-	Surgery	Surgery	6
Ghazi AA et al., 2011 [33]	1	F, 35	Pancreatic	G1-2, Ki67 = 1–3%	IIIA	-	At diagnosis	PTHrP	-	IV hydration Biphosphonates Calcitonin	Surgery Etoposide Platinum	Biphosphonates Surgery	6
Shirai K et al., 2011 [34]	1	F, 53	Pancreatic	NA	IIIA	-	At diagnosis	PTHrP	Glucagon	IV hydration	Surgery HACE RFA	Surgery	84
Takeda K et al., 2010 [35]	1	M, 12	Pheochromocytoma	NA	NA	-	At diagnosis	PTHrP	-	Biphosphonates Loop diuretics	Surgery	Surgery	12
Morita Y et al., 2010 [36]	1	F, 58	Pancreatic	NA	IV	Liver	At diagnosis	PTHrP	Gastrin	-	Surgery	Surgery	19
Demura M et al., 2010 [37]	1	F, 20	MTC	NA	NA	Lymphnodes	At diagnosis	PTH	Calcitonin	-	Surgery	Surgery	NA
Srirajaskanthan R et al., 2009 [38]	5	F, 25	Pancreatic	NA	NA	-	At diagnosis	PTHrP	-	NA	SSA TAE Surgery OLT	Surgery	NA
		F, 44	Pancreatic	NA	IV	Liver	At diagnosis	PTHrP	-	NA	SSA Streptozotocine 5-FU	SSA	NA
		M, 26	Pancreatic	NA	IV	Liver	At diagnosis	PTHrP	-	NA	SSA Streptozotocine 5-FU Cisplatin Etoposide		NA
		F, 64	Pancreatic	NA	NA	-	At diagnosis	PTHrP	-	NA	SSA Streptozotocine 5-FU Cisplatin Etoposide		NA
		F, 34	Pancreatic	NA	IV	Liver	At diagnosis	PTHrP	-	NA	SSA Streptozotocine 5-FU Cisplatin Carboplatin Etoposide Surgery TAE	Surgery	NA
Brzozowska MM et al., 2009 [39]	1	F, 77	Unknown	NA	IV	Liver Spleen	At diagnosis	PTHrP	-	Biphosphonates Corticosteroids	Etoposide Carboplatin SSA	-	14
Van den Eynden GG et al., 2007 [40]	1	M, 59	Pancreatic	G1	IIIA	-	At diagnosis	PTHrP	Calcitonin	Biphosphonates IV hydration	Surgery SSA IFNα	IFNα	57
Barakat MT et al., 2004 [41]	1	F, 47	Pancreatic	NA	IV	Liver	24	PTHrP	-	Biphosphonates IV hydration	TAE SSA	TAE SSA	40
Mullerpatan PM et al., 2004 [42]	1	F, 56	Pancreatic	NA	IIB	-	At diagnosis	NA	Calcitonin VIP	IV hydration	Surgery	IV hydration	18
Abraham P et al., 2002 [43]	1	F, 25	Pancreatic	NA	NA	-	At diagnosis	PTHrP	-	Biphosphonates	Surgery	Biphosphonates Surgery	24
Clemens P et al., 2001 [44]	1	M, 34	Pancreatic	NA	IIIA	-	At diagnosis	PTHrP	-	IV hydration Biphosphonates Calcitonin Corticosteroids	Streptozotocin 5-FU Doxorubicin SSA Carboplatin Etoposide	Chemotherapy	32, DOD
Papazachariou IM et al., 2001 [45]	2	F, 33	Pancreatic	NA	IV	Liver	At diagnosis	PTHrP	Somatostatin	Biphosphonates IV hydration	Surgery TAE SSA	Surgery TAE	60
		M, 41	Pancreatic	NA	IV	Liver	48	PTHrP	Somatostatin	-	TAE Surgery	TAE Surgery	4, DOD
Loh K et al., 1998 [46]	1	M, 15	Retroperitoneal paraganglioma	-	IV	Liver Bone Mediastinum	At diagnosis	PTHrP	-	Biphosphonates Calcitonin IV hydration	Surgery	Biphosphonates Surgery	4
van de Loosdrecht AA et al., 1998 [47]	1	F, 45	Pancreatic	NA	IV	Liver	128	PTHrP	-	IV hydration Corticosteroids	SSA	-	16, DOD
Mantzoros CS et al., 1997 [48]	1	F, 59	Unknown	NA	IV	Liver	At diagnosis	PTHrP	-	IV hydration Biphosphonates Calcitonin Plicamycin Gallium nitrate	5-FU Carboplatin	-	3, DOD
Wu TJ et al., 1997 [49]	9	M, 66	Pancreatic	NA	IV	Liver Spleen	NA	NA	-	NA	NA	NA	NA
		F, 42	Pancreatic	NA	IV	Liver Spleen	NA	PTHrP	-	NA	NA	NA	NA
		F, 45	Pancreatic	NA	IV	Liver	NA	PTHrP	-	NA	NA	NA	NA
		M, 64	Pancreatic	NA	IV	Liver	NA	PTHrP	Glucagon	NA	NA	NA	NA
		M, 61	Pancreatic	NA	NA	-	NA	PTHrP	-	NA	NA	NA	NA
		F, 38	Pancreatic	NA	IV	Liver	NA	PTHrP	-	NA	NA	NA	NA
		M, 20	Pancreatic	NA	NA	-	NA	PTHrP	-	NA	NA	NA	NA
		F, 47	Pancreatic	NA	IV	Liver	NA	PTHrP	-	NA	NA	NA	NA
		F, 51	Pancreatic	NA	IV	Liver	NA	PTHrP	Somatostatin	NA	NA	NA	NA
Mao C et al., 1995 [50]	3	M, 41	Pancreatic	NA	IV	Liver Lymphnodes	0.5	NA	-	IV hydration Calcitonin Plicamycin	Surgery	-	32, DOD
		M, 43	Pancreatic	NA	IV	Liver	120	PTHrP	-	-	Surgery Chemotherapy	-	6, DOD
		M, 64	Pancreatic	NA	IV	Liver Lymphnodes Kidney Pleurae	At diagnosis	PTHrP	-	Corticosteroids	-	-	1.5, DOD
Anthony LB et al., 1995 [51]	1	F, 75	Pancreatic	NA	NA	-	60	PTHrP	PP	IV hydration Plicamycin	Streptzotocin 5-FU SSA	SSA	3
Ratcliffe WA et al. [52]	1	F, 39	Pancreatic	NA	NA	-	At diagnosis	PTHrP	-	IV hydration Calcitonin Biphosphonates	Surgery	Surgery	9
Yoshikawa T et al., 1994 [53]	1	M, 43	Thymic	NA	NA	-	At diagnosis	PTH (serum) PTHrP (immunohistochemistry)	-	-	RT	-	15
Mune T et al., 1993 [54]	1	M, 58	Pheochromocytoma	NA	NA	-	At diagnosis	PTHrP	-	Alpha-blockers	Surgery	Alpha-blockers	NA
Williams EJ et al., 1992 [55]	1	M, 30	Pancreatic	NA	IIIA	-	At diagnosis	PTHrP	Somatostatin PP	Biphosphonates IV hydration Calcitonin Plicamycin	Streptozotocin	Plicamycin Streptozotocin	23, DOD
Bridgewater JA et al., 1993 [56]	1	M, 68	Pheochromocytoma	NA	III	Lymphonodes	94	PTHrP	-	Biphosphonates	Surgery	Biphosphonates	1, DOD
Miraliakbari BA et al., 1992 [57]	1	F, 47	Pancreatic	NA	IIIA	-	At diagnosis	PTHrP	-	IV hydration Calcitonin Corticosteroids	Surgery	Surgery	36
Tarver DS et al., 1992 [58]	1	M, 36	Pancreatic	NA	IV	Liver	48	PTHrP	-	IV hydration Calcitonin Biphosphonates Corticosteroids	TAE	TAE	18
Mitlak BH et al., 1991 [59]	1	F, 77	Pancreatic	NA	NA	-	At diagnosis	PTHrP	-	Biphosphonates	Surgery Streptozotocin 5-FU	Surgery Biphosphonates Streptozotocin 5-FU	58
Bresler L et al., 1991 [60]	1	M, 45	Pancreatic	NA	NA	-	At diagnosis	NA ^a^	-	NA	Surgery Streptozotocin 5-FU	Surgery	60
Harrison M et al., 1990 [61]	1	M, 51	Pheochromocytoma	NA	NA	-	At diagnosis	PTHrP	-	Biphosphonates IV hydration	Surgery Cisplatin Doxorubicin Metotrexate 5-FU Lomustine SSA	Surgery SSA	49
Kimura S et al., 1990 [62]	1	M, 54	Pheochromocytoma	NA	NA	-	At diagnosis	PTHrP	-	-	Surgery	Surgery	NA
Rizzoli R et al., 1990 [63]	2	M, 30	Pancreatic	NA	IV	Liver	At diagnosis	PTHrP	-	Bisphosphonates	Surgery	Surgery	NA
		F, 60	Pancreatic	NA	IV	Liver Bone	At diagnosis	PTHrP	-	Bisphosphonates	IFNα	Bisphosphonates	12, DOD
Dodwell D et al., 1990 [64]	1	F, 42	Pancreatic	NA	IV	Liver	At diagnosis	PTHrP	-	Corticosteroids Bisphosphonates	IFNα SSA Surgery	SSA Surgery	30
Wynick D et al., 1990 [65]	1	F, 37	Pancreatic	NA	IV	Liver	At diagnosis	PTHrP	-	Corticosteroids	SSA	SSA	48
Heitz PU et al., 1989 [66]	1	F, 52	Pancreatic	NA	NA	NA	NA	NA	NA	NA	NA	NA	NA
Venkatesh S et al., 1989 [67]	1	M, 54	Pancreatic	NA	IV	Liver	72	NA	VIP	IV hydration	TAE Surgery SSA	IV hydration SSA	48
Friesen SR, 1987 [68]	1	M, 8	Pancreatic	NA	IV	Liver	At diagnosis	NA ^a^		Phosphate enemas	Surgery	Phosphate enemas Surgery	2, DOD
Sarfati E et al., 1987 [69]	1	M, 64	Pulmonary	NA	NA	-	At diagnosis	NA	-	-	Surgery	Surgery	18
Shetty MR, 1987 [70]	1	M, 44	Pancreatic	NA	IV	Liver	NA	NA	NA	NA	NA	NA	NA
Vair DB et al., 1987 [71]	1	F, 47	Pancreatic	NA	NA	-	At diagnosis	NA ^a^	-	NA	NA	NA	NA
Arps H et al., 1986 [72]	1	M, 48	Pancreatic	NA	IV	Liver	76	PTH	-	NA	Surgery TAE	-	7
Grossman E et al., 1985 [73]	4	M, 16	Pheochromocytoma	NA	NA	NA	At diagnosis	NA	-	NA	Surgery	Surgery	NA
		F, 75	Pheochromocytoma	NA	NA	NA	At diagnosis	NA	-	NA	Surgery	NA	NA
		M, 26	Pheochromocytoma	NA	NA	NA	At diagnosis	NA	-	NA	Surgery	Surgery	NA
		M, 58	Pheochromocytoma	NA	NA	NA	At diagnosis	NA	-	NA	Surgery	NA	NA
Baba T et al., 1985 [74]	1	M, 15	Pheochromocytoma	NA	IV	Bone	204	NA	-	IV hydration Calcitonin Plicamycin	Surgery	Plicamycin	3, DOD
Loveridge N et al., 1985 [75]	1	F, 68	Pulmonary	NA	IV	Liver	At diagnosis	NA ^a^	-	Bisphosphonates IV hydration Corticosteroids	-	IV hydration	12, DOD
Shanberg AM et al., 1985 [76]	1	M, 53	Pheochromocytoma	NA	NA	-	At diagnosis	NA	-	IV hydration Corticosteroids	Surgery	IV hydration Surgery	NA
Stewart AF et al., 1985 [77]	1	F, 11	Pheochromocytoma	NA	NA	-	At diagnosis	NA ^a^	-	-	Surgery	Surgery	NA
Rasbach DA et al., 1985 [78]	1	F, 68	Pancreatic	NA	IV	Liver	At diagnosis	NA ^a^	-	IV hydration Prednisone	Streptozotocine 5-FU	Streptozotocine 5-FU	3
Fairhurst JB et al., 1981 [79]	1	M, 47	Pheochromocytoma	NA	NA	-	At diagnosis	NA ^a^	-	-	Surgery	Surgery	9
Öberg K et al., 1981 [80]	3	M, 52	Pancreatic	NA	NA	-	At diagnosis	NA	Calcitonin VIP Gastrin PP	IV hydration	Surgery Streptozotocin	Surgery	20
		M, 38	Pancreatic	NA	IV	Liver Omentum	94	NA	Calcitonin VIP	IV hydration	Surgery Streptozotocin	Surgery	7
		F, 54	Pancreatic	NA	IV	Omentum	At diagnosis	NA	Calcitonin PP	IV hydration	Surgery Streptozotocin	Streptozotocin	6
De Plaen JF et al., 1976 [81]	1	M, 45	Pheochromocytoma	NA	NA	-	At diagnosis	NA	-	-	Surgery	Surgery	1, DOD
Ghose RR et al., 1976 [82]	1	M, 14	Pheochromocyotma	NA	NA	-	At diagnosis	PTH	-	-	Surgery	Surgery	NA
Gray RS et al., 1976 [83]	1	M, 66	Pheochromocytoma	NA	NA	-	At diagnosis	NA	-	-	Surgery	Surgery	18
Cryer PE et al., 1976 [84]	1	F, 61	Pancreatic	NA	IV	Liver	189	NA	Gastrin	Calcitonin Plicamycin	Streptozotocin	Streptozotocin	13
Deftos LJ et al., 1976 [85]	1	F, 27	Gastric	NA	IV	Liver	At diagnosis	PTH	Calcitonin	-	Melphalan	-	18, DOD
Hirose S et al., 1975 [86]	1	M, 62	Pancreatic	NA	IV	Liver	At diagnosis	PTH	^- b^	IV hydration	-	-	4, DOD
Kukreja SC et al., 1973 [87]	1	M, 16	Pheochromocytoma	NA	NA	-	At diagnosis	PTH	-	Diuretics	Surgery	Surgery	36
DeWys WD et al., 1973 [88]	1	M, 57	Pancreatic	NA	IV	Liver	At diagnosis	NA ^a^	ACTH	IV hydration Calcitonin	Streptozotocin	Streptozotocin	14
Swinton NW et al., 1972 [89]	1	M, 12	Pheochromocytoma	NA	NA	-	At diagnosis	NA	-	-	Surgery	Surgery	25
Lopes VM et al., 1970 [90]	1	F, 42	Pancreatic	NA	NA	-	At diagnosis	NA	^- b^	IV hydration	Surgery	Surgery	24
Murray JS et al., 1961 [91]	1	M, 49	Pancreatic	NA	NA	-	At diagnosis	NA	^- b^	IV hydration	Surgery	-	14

^a^ Evidence for secretion of PTH-like substance (probably PTHrP); ^b^ Probably VIP. Abbreviations: 5-FU, 5-fluorouracil; CapTem, Capecitabine + Temozolomide; DOD, died of disease; ENETS, European Neuroendocrine Tumor Society; F, female; HACE, hepatic artery chemoembolization; IFNα, Interferon-alpha; IV, intravenous; M, male; MTC, medullary thyroid cancer; NA, not available; NEN, neuroendocrine neoplasia; OLT, orthotopic liver transplantation; PP, pancreatic polypeptide; PRRT, peptide receptor radionuclide therapy; PTH, parathyroid hormone; PTHrP, parathyroid hormone-related peptide; RT, radiotherapy; SSA, somatostatin analogue; TAE, transarterial embolization; VIP, vasoactive intestinal peptide.

**Table 2 jpm-12-01553-t002:** Demographic, pathological, and clinical characteristics of well-differentiated NEN patients with paraneoplastic hypercalcemia reported in the literature.

Total Number of Cases	114 (100%)
**Sex**	*n* = 113
Male	62 (54.9%)
Female	51 (45.1%)
**Mean age ± standard deviation**	46.3 ± 15.8
**Primary NEN histology**	*n* = 114
Pancreatic NEN	83 (72.8%)
Pheochromocytoma	18 (15.8%)
Unknown NEN	4 (3.5%)
Lung NEN	3 (2.6%)
Gastric NEN	1 (0.9%)
Neck NEN	1 (0.9%)
Mediastinal NEN	1 (0.9%)
Medullary thyroid cancer	1 (0.9%)
Paraganglioma	1 (0.9%)
Thymic NEN	1 (0.9%)
**Tumor grade**	*n* = 23
G1	10 (43.5%)
G2	12 (52.2%)
G2	1 (4.3%)
**Metastatic disease at paraneoplastic hypercalcemia onset**	66 (57.9%)
**Metastatic sites**	*n* = 110
Liver	65 (59.1%)
Lymphnode	9 (8.2%)
Bone	6 (5.5%)
Lung	1 (0.9%)
Other sites	12 (10.1%)
**Presence of paraneoplastic hypercalcemia at NEN diagnosis**	79 (69.3%)
**Mean time to onset of paraneoplastic hypercalcemia, months ± standard deviation**	83.4 ± 56.3
**Causes of metachronous paraneoplastic hypercalcemia**	*n* = 16
Local recurrence/development of distant metastases	7 (43.8%)
Tumor progression	8 (50%)
No disease progression	1 (6.3%)
**Calcemic levels at onset of paraneoplastic hypercalcemia, mean ± SD (mg/dl)**	14 ± 2.7
Calcemic levels at onset of paraneoplastic hypercalcemia in pancreatic NEN patients	14.3 ± 2.9
Calcemic levels at onset of paraneoplastic hypercalcemia in Pheochromocytoma patients	12.4 ± 1.4
**Paraneoplastic hypercalcemia-producing molecules**	*n* = 80
PTHrP	68 (85%)
PTH	9 (11.3%)
1,25(OH) vitamin D	3 (3.8%)
**Cosecretion of other peptides**	32 (28.1%)
Calcitonin	10 (31.3%)
VIP	8 (25%)
Pancreatic polypeptide	6 (18.8%)
Gastrin	5 (15.6%)
Somatostatin	5 (15.6%)
Glucagon	4 (12.5%)
ACTH	1 (3.1%)
Cosecretion of more than one peptide	7 (21.9%)
**Median survival, months (IQR range)**	18 months (range, 7–37)

Abbreviations: ACTH, adrenocorticotropic hormone; IQR, interquartile range; NEN, neuroendocrine neoplasms; PTH, parathyroid hormone; PTHrP, PTH-related peptide; VIP, vasoactive intestinal peptide.

## Data Availability

Not applicable.

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
