# Peer review of "What Lies behind Paraneoplastic Hypercalcemia Secondary to Well-Differentiated Neuroendocrine Neoplasms? A Systematic Review of the Literature"

_jpm, 2022, doi:10.3390/jpm12101553_

Round 1
Reviewer 1 Report
The main question of this review was to address the timing and prevalence of hypercalcemia in well-differentiated neuroendocrine neoplasms. Due to the paucity of reviews, it was relevant to systematize information presented in the current medical literature.The review is well-structured and easy to read. The presented tables are well organized and informative.
Conclusions are consistent with the text of the manuscript . Probably, the table which addresses the differences between hypercalcemia in paraneoplastic syndrome related to solid and hematologic malignancies vs hypercalcemia in neuroendocrine tumors would be added.
The better prognosis of patients with paraneoplastic hypercalcemia in well-differentiated NEN is probably related to a lower Ki-index, thus the more indolent course of the disease is related to better survival.
Author Response
we thank the reviewer for its comments and suggestions that help us to improve our MS. We provided in the text a new paragraph to summarize the most relevant differences between paraneoplastic hypercalcemia in NENs, solid and hematological tumor. Section: Discussion, Page 5 lines 351-358.

Reviewer 2 Report
This review is a comprehensive analysis from multiple previous literatures. The authors have done a good analysis and the conclusion they have made are very clear.
Author Response
We thank the reviewer for his comments, we are pleased with his appreciation for our work and we are pleased to submit the revised version of the MS.
